# Immunomodulation of Oxidative Stress during Organ Donation Process: Preliminary Results

**DOI:** 10.3390/healthcare10050762

**Published:** 2022-04-20

**Authors:** Nora Palomo-López, Ana Rodríguez-Rodríguez, Luis Martín-Villén, María Mendoza-Prieto, Zaida Ruiz de Azúa-López, Lluis Sempere-Bordes, Laura Boyero-Corral, Domingo Daga-Ruiz, Antonio Gordillo-Brenes, María Pacheco-Sánchez, José Miguel Perez-Villares, Ángel Vilches-Arenas, Juan José Egea-Guerrero

**Affiliations:** 1Department of Critical Care, Virgen del Rocío University Hospital, Av. Manuel Siurot s/n, 41013 Seville, Spain; norapalomolpz@gmail.com (N.P.-L.); luis.martin.villen.sspa@juntadeandalucia.es (L.M.-V.); mmp_1209@hotmail.com (M.M.-P.); zaida.ruizazua.sspa@juntadeandalucia.es (Z.R.d.A.-L.); juanj.egea.sspa@juntadeandalucia.es (J.J.E.-G.); 2Biomedical Institute of Seville, IBiS/CSIC/University of Seville, Calle Antonio Maura Montaner, 41013 Seville, Spain; ana.rodriguez.r.sspa@juntadeandalucia.es (A.R.-R.); luissempere@yahoo.es (L.S.-B.); lboyero@hotmail.com (L.B.-C.); 3Department of Critical Care, Virgen de la Victoria University Hospital, Campus de Teatinos s/n, 29010 Málaga, Spain; domingo.daga@gmail.com; 4Department of Critical Care, Puerta del Mar University Hospital, Av. Ana de Viya, 21, 11009 Cádiz, Spain; angobre@gmail.com; 5Seville-Huelva Transplant Coordination Center, Av. Manuel Siurot s/n, 41013 Sevilla, Spain; maria.pacheco.sanchez.sspa@juntadeandalucia.es; 6Department of Critical Care, Virgen de las Nieves University Hospital, Av. de las Fuerzas Armadas, 2, 18014 Granada, Spain; josem.perez.villares.sspa@juntadeandalucia.es; 7Department of Preventive Medicine, University of Seville, Av. Sánchez Pizjuán s/n, 41009 Seville, Spain

**Keywords:** brain death donor, donation after circulatory death, ischemia, oxidative stress, malondialdehyde, melatonin, antioxidant effect

## Abstract

The objective was to quantify oxidative stress resulting from ischemia during the donation process, using malondialdehyde (MDA) measurement, and its modulation by the administration of melatonin. We designed a triple-blind clinical trial with donors randomized to melatonin or placebo. We collected donors by donation after brain death (DBD) and controlled donation after circulatory death (DCD), the latter maintained by normothermic regional perfusion (NRP). Melatonin or placebo was administered prior to donation or following limitation of therapeutic effort (LTE). Demographic variables and medical history were collected. We also collected serial measurements of MDA, at 60 and 90 min after melatonin or placebo administration. A total of 53 donors were included (32 from DBD and 21 from DCD). In the DBD group, 17 donors received melatonin, and 15 placebo. Eight DCD donors were randomized to melatonin and 13 to placebo. Medical history and cause for LTE were similar between groups. Although MDA values did not differ in the DBD group, statistical differences were observed in DCD donors during the 0–60 min interval: −4.296 (−6.752; −2.336) in the melatonin group and −1.612 (−2.886; −0.7445) in controls. Given the antioxidant effect of melatonin, its use could reduce the production of oxidative stress in controlled DCD.

## 1. Introduction

Solid organ transplantation is often the only option for patients with end-stage diseases [1]. Higher life expectancy and improved therapeutic techniques have added to an upsurge of patients on transplant waiting lists. This fact, coupled with a decrease in donation after brain death (DBD), makes the available organ supply insufficient [2]. The resulting imbalance between organ supply and demand has forced the Spanish National Transplant Organization to maximize organ donation strategies, including controlled donation after circulatory death (cDCD) [3]. Unfortunately, the ischemia that occurs during DCD is inherent to this donation process, exposing grafts to hypoxia and hypoperfusion, and leading to poorer outcomes in liver and kidney transplants compared with DBD, in which the ischemia is less relevant [4,5]. Current efforts to improve DCD outcomes, and make them comparable with those from DBD, focus on limiting the negative consequences of ischemia [6]. 

One of the deleterious phenomena triggered by ischemia is oxidative stress, which arises from an imbalance between free radical production and antioxidant defenses. These free radicals, a product of oxygen metabolism, make up reactive oxygen species (ROS), along with other molecules. ROS react with carbohydrates, proteins, nucleic acids, and lipids. Lipid peroxidation (LPO), the specific reaction between ROS and lipids, causes cell damage and eventually, cell death or apoptosis (Figure 1) [7]. During LPO, various molecules are generated and released into the bloodstream, such as α, β-unsaturated aldehydes (4-HNE and acrolein), di-aldehydes (including malondialdehyde (MDA)), and keto-aldehydes (4-oxo-trans-2-nonenal (ONE) and isoketals) [8]. Due to its properties, MDA has proved to be the most reliable molecule to measure. The release of MDA into the bloodstream allows its determination in the laboratory, thus it has been postulated that it could allow the quantification of oxidative stress [7]. 

Different techniques have been used to reduce the impact of ischemic time during organ donation, such as donor maintenance by normothermic regional perfusion (NRP) or the use of ex vivo devices, among others. These procedures attempt to counteract the deleterious effects of oxidative stress by providing oxygen and therapeutic measures to treat metabolic acidosis and lacticaemia [9,10]. However, these procedures alone do not offset the lethal effects of ischemic oxidative stress, particularly in DCD. As the transplant community increasingly relies on DCD to increase the organ pool, more efforts are urgently needed to quantify and debilitate oxidative stress and its destructive activity in the donation process. 

Melatonin is a highly lipophilic molecule, endogenously synthesized in the pineal gland, retina, and digestive tract. As well as its role in the regulation of the sleep–wake cycle and circadian rhythms being well known, it has other noteworthy functions. It provides protection against certain types of cancer, attenuates neurodegenerative diseases including Alzheimer’s, and functions as an immunomodulator and antioxidant [11]. This last property results from its ability to decrease nitric oxide synthesis, lower cyclooxygenase-2 levels, and reduce the production of inflammatory cytokines. These changes stimulate the production of anti-inflammatory cytokines, which reduces mitochondrial damage and cell death. Melatonin has, therefore, an ability to modulate oxidative stress (Figure 2) [12].

The present study aims to quantify oxidative stress, in both DBD and controlled DCD, by means of MDA measurement and the temporal assessment of its evolution after melatonin administration. 

## 2. Materials and Methods

We conducted a randomized, multicenter, triple-blind clinical trial that included DBD and DCD donors from March 2018 to December 2019 (ISRCTN Registry: Study ID ISRCTN66157570). The protocol was approved by the CEI of the Hospitales Universitarios Vírgen Macarena-Virgen del Rocío Review Board, Seville, Spain. Date: 3 December 2018 (Acta11/2018). Ref: 1013-N17.

DBD was defined as the irreversible loss of all brain functions, including the brain stem. The presence of either brain death or a catastrophic and irreversible brain injury leads to fulfilling the brain death criteria. DCD was defined as the retrieval of organs for the purpose of transplantation from patients whose death is diagnosed and confirmed using cardio-circulatory criteria. 

Patients were selected from donor hospitals authorized by the Transplant Coordination Network in Andalusia, Spain. Each donor set was randomly divided into two groups: melatonin and controls. A list was generated by an electronic system (N-Qery advisor) with the randomization codes. Donors received melatonin or placebo in accordance with inclusion criteria and their randomized code.

Melatonin or placebo was administered via nasogastric tube at the time of determination of death by either neurological or circulatory criteria. The melatonin group received 30 mg of melatonin diluted in 20 cc of sucrose solution (0.4 g/dl). Controls received 20 cc of diluted sucrose solution. Dosage selection was based on the positive results from trials by Mistraletti et al. and Dianatkhah et al. on critically ill patients [13,14]. 

Medical history, demographic variables, and cause of BD were collected. Blood extractions were performed at 0’, 60’, and 90’ after melatonin/placebo administration. In each extraction, a Vacutainer EDTA K3 tube was collected for MDA determination in blood plasma. Differences in MDA concentration were calculated for 0–60’ and 0–90’ intervals.

DCD donors were maintained using NRP, in accordance with Spain’s National Transplant Organization protocol [15]. Femoral cannulation for arterial perfusion and venous drainage were initiated concurrently with clamping of the supraceliac aorta (Fogarty Clamp). Once aortic clamping was secured, NRP and maintenance commenced. All necessary adjustments were made to maintain the physiological parameters required for the correct perfusion of abdominal organs [15]. 

Two ischemic intervals were discerned in the DCD group: functional warm ischemia time (f-WIT), defined as the interval from appearance of systolic blood pressure at <60 mmHg to initiation of preservation, and total warm ischemia time (TIT), defined as the interval from withdrawal of life support to initiation of preservation [15].

### 2.1. Quantification of Malondialdehyde

Quantification of MDA in plasma was performed using an absorbance-based technique. MDA reacts with thiobarbituric acid, forming a complex known as “thiobarbituric acid reactive substances” or TBARS, which can be measured by spectrophotometry or fluorimetry.

### 2.2. Statistical Analysis

A descriptive analysis of the sample was performed. Qualitative data were presented as frequencies and percentages. Quantitative variables were expressed as means and standard deviation (SD) or median and interquartile range (25–75) (IR), based on their distribution.

For inferential analysis, a Chi-Square Test was used to compare qualitative variables and the Mann–Whitney U test was used for quantitative data. For statistical estimates, 95% confidence intervals (CIs) were calculated, and *p* < 0.05 was considered statistically significant. Statistical analysis was performed using SPSS V25 software (IBM SPSS Statistics for Windows, Version 25.0. Armonk, NY, USA: IBM Corp.).

## 3. Results

The sample included a total of 53 donors (32 DBD and 21 DCD). In the DBD group, 17 were randomized to melatonin and 15 to placebo. Of the 21 DCD donors, 8 received melatonin and 13 were given placebo (Figure 3). 

Table 1 displays demographic variables, medical history, and cause of brain injury for patients that progressed to DBD. Different distributions in rate of alcoholism were observed between melatonin-treated vs. non-treated groups, and the most frequent cause for brain injury in both groups was hemorrhagic cerebrovascular accident. Median age was 56 (IR: 42.3–64.6) for melatonin recipients and 63 (RI: 46.6–69.9) for controls. The mean time for progression to BD for both groups was two days.

Table 2 provides demographic data, medical history, and cause for limitation of therapeutic effort (LTE) for DCD donors. No differences were found between melatonin recipients and controls. The most frequent cause for LTE in both groups was anoxic encephalopathy. The median age for melatonin-treated donors was 61.5 (IR: 53.5–67.7), and 61.0 (IR: 53–68.5) for placebo recipients. Analyses of functional WIT, total WIT, and duration of NRP revealed similar distributions for both groups.

Table 3 illustrates data on MDA measurement in DBD donors. No significant differences were found in MDA concentration, in either melatonin or placebo recipients. These results contrast with those of DCD donors (Table 4) which showed a decrease in MDA values at 60’ and 90’ after melatonin/placebo administration (Table 4). The most notable differences were observed at 90’, with MDA concentrations of −4.3 (−6.7; −2.6) in melatonin recipients, and −1.6 (−2.9; −0.75) in controls.

## 4. Discussion

The present work shows a reduction in oxidative stress, expressed by a decrease in MDA concentration in DCDs treated with melatonin. During the DBD process, organs undergo cold perfusion before organ retrieval, which minimalizes WIT and reduces ischemic injury. Additionally, during DBD the organs suffers less from ischemia, compared with DCD. Therefore, we postulate that MDA production will be lower, which would diminish the impact of oxidative stress on DBD donors. This would explain why melatonin administration, as well as the attempt to reverse oxidative stress, did not show any improvement in the DBD group.

Our results show different alcoholism rates between the melatonin and placebo groups in DBD; however, the oxidative stress level was similar in DBD. There was no other difference between the melatonin and placebo groups in the donor characteristics. 

The efficacy of melatonin as an antioxidant has been widely reported in experimental studies on cardiac ischemia, traumatic brain injury (TBI), tumor pathology, and chronic disease, such as pulmonary fibrosis or diabetes [16]. In an experimental animal study, Lee and colleagues noted that cases which received melatonin after treatment with mesenchymal stem cells obtained a reduction in ischemic stress conditions [17]. Similarly, Li and coauthors indicated that melatonin may improve left ventricular function in cardiac cells undergoing hypoxia/reperfusion processes after acute myocardial infarction, thereby decreasing the extent of the infarction while lowering the number of myocytes affected by ischemia [18].

In the field of organ donation, several experimental animal models have also demonstrated the benefits of melatonin [6]. Aslaner and colleagues maintained renal grafts in perfusion liquid mixed with melatonin during cold ischemia. They observed decreased levels of MDA and lactate dehydrogenase (LDH), and an overall reduction in graft histological damage as compared with implants that were maintained without melatonin [6]. Similarly, studies on animal model kidney transplants showed reduced histological damage and improved graft survival in cases where melatonin was administered prior to transplantation [18]. Baykaraand and colleagues used histological sampling, after a provoked hour-long hepatic ischemia, to compare apoptotic cell count in melatonin-treated vs. non-treated groups. The authors reported a drop in the rate of cell death in melatonin recipients [19]. 

Lan and coauthors researched the efficacy of melatonin in a DCD animal model after heart transplantation. In this study, donors received melatonin treatment for one week and 30 min prior to donation. After heart graft extraction, ex vivo techniques were applied for another 30 min prior to transplantation. Oxidative stress was measured by MDA determination at 3 h post-implantation. Histological and anthropometric studies of the grafts were also carried out. The authors concluded that grafts previously treated with melatonin showed fewer edemas and less inflammation, in addition to decreased MDA values, indicative of reduced oxidative stress. Interestingly, a decrease in apoptotic myocytes was also observed in the experimental group [20]. In summary, these studies confirm the beneficial effects of melatonin treatment, highlighting this molecule’s capacity to reduce damage caused by ischemia-reperfusion. Nevertheless, these works were carried out in animal models, and human data to corroborate these findings is still lacking.

We identified the following limitations in our study. First, we did not track MDA levels after the 0–90 min interval. However, it is our belief that the during the DCD process, most alterations in the internal environment occur during the initial moments after maintenance, with organ viability and procedures usually adjusted to this initial stage of NRP as well. Secondly, there is an absence of corroborative results in terms of graft functionality, and thus we cannot confirm that a change in oxidative stress leads directly to a better outcome in the subsequent transplant. As these are preliminary results, we hope to be able to provide an appropriate response to this question. Thirdly, despite differences in DBD rates of alcoholism, a background which is known to be associated with oxidative stress, no differences were observed in oxidative stress levels [21]. Finally, the DCD donor sample is small, which could decrease the validity of our results.

Nevertheless, to our knowledge, this is the first multicenter trial in the underexplored field of controlled DCD, which identifies the positive potential of melatonin use at an experimental level. If confirmed, it could provide a substantial boost to controlled DCD organ management, by improving outcome and decreasing the harmful effects of ischemia. 

## 5. Conclusions

If the donor organ shortage is to be resolved, the transplant community must increasingly rely on deceased organ donation, including controlled DCD and DBD. Our preliminary data suggest that the administration of melatonin might reduce MDA levels in DCD donors, and probably diminish oxidative stress and its deleterious effects in this type of donation. Despite this, there is scarce research to corroborate the possible benefits of this treatment on transplanted grafts. Our findings open up new possibilities and hypotheses for melatonin use development in these donor populations. More studies are mandatory to corroborate our results and to confirm improved outcomes in donation-transplantation programs through these strategies. 

## Figures and Tables

**Figure 1 healthcare-10-00762-f001:**
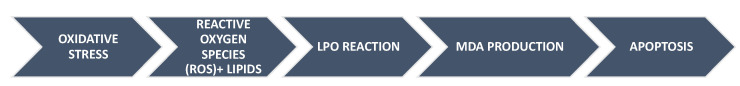
Representation of oxidative stress cascade and its effects.LPO: lipid perodixation. ROS: reactive oxygen species. MDA: Malondialdehyde.

**Figure 2 healthcare-10-00762-f002:**
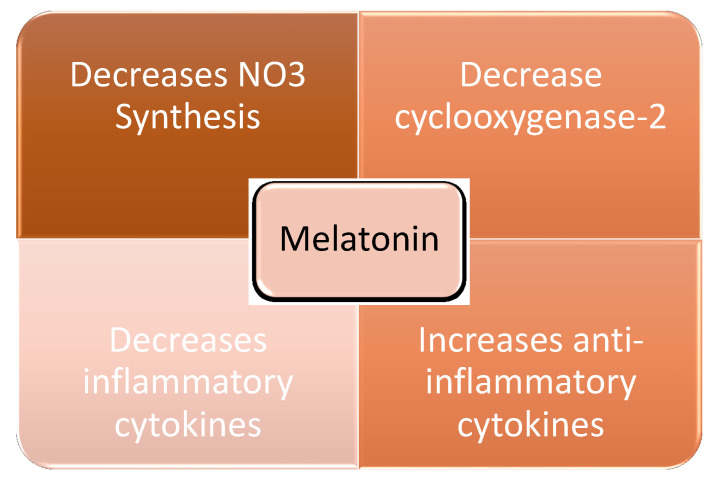
Melatonin properties which control oxidative stress. NO3: Nitrate.

**Figure 3 healthcare-10-00762-f003:**
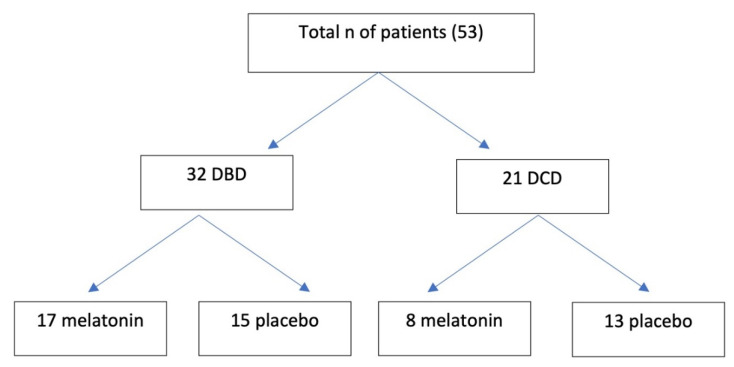
Distribution of the patients included in the study. n: number; DBD: Donation after brain death; DCD: donation after circulatory death.

**Table 1 healthcare-10-00762-t001:** Clinical variables of donation after brain death donors group.

Variables		Melatonin Group	Placebo Group	*p*
**Age**		56 (42.3–64.7)	63 (46.6–69.9)	0.75
Gender, male		10 (58.8)	9 (60)	0.62
Type of brain injury	Hemorrhagic CVA	7 (41.1)	7 (40.8)	0.41
	TBI	6 (35.3)	4 (26.7)	
	Ischemic CVA	71 (5.9)	4 (26.7)	
	Anoxic encephalopathy	3 (17.6)	1 (6.7)	
Smoker		7 (41.2)	4 (28.6)	0.36
Alcoholism		7 (41.2)	1 (7.1)	0.04
AH		7 (41.2)	8 (57.1)	0.30
DM		3 (17.6)	2 (14.3)	0.59
ASA		2 (11.8)	2 (11.4)	0.62
OAC		3 (17.6)	1 (7.1)	0.38
GCS at admission		6 (3.9–7.9)	3 (2.9–7.9)	1
Diabetes Insipidus		12 (70.6)	7 (46.7)	0.16
Length of stay in ICU		2 (1.9–9.9)	1 (0.84–3.3)	0.47
Length of stay in Hospital		2 (1.9–9.9)	2 (1.39–4)	0.50

Clinical variables of patients enrolled in the clinical trial who progressed to donation after brain death (DBD) and were organ and tissue donors. Qualitative variables were expressed as frequencies (%) and quantitative variables as median and interquartile range (25–75). IR, Interquartile Range; TBI, Traumatic Brain Injury; CVA, Cerebrovascular Accident; AH, Arterial Hypertension; DM, Diabetes Mellitus; ASA, Aspirin; OAC, Oral anticoagulation; GCS, Glasgow Coma Score; ICU, Intensive Care Unit.

**Table 2 healthcare-10-00762-t002:** Clinical variables of the donation after cardiac death group.

	Melatonin Group	Placebo Group	*p*
Age, median (RI)	61.5 (53.5–67.8)	61 (53–8.5)	0.75
Gender, male *n* (%)	6 (75)	7 (53.8)	0.40
Smoker *n* (%)	4 (32.8)	5 (62.5)	0.16
Alcoholism *n* (%)	3 (23.1)	3 (37.5)	0.13
AH *n* (%)	11 (84.6)	7 (87.5)	0.92
DM *n* (%)	4 (30.8)	2 (25)	0.78
Liver disease *n* (%)	2 (15.4)	2 (25)	0.59
Kidney disease *n* (%)	0 (0)	1 (12.5)	0.20
Heart disease *n* (%)	4 (30.8)	3 (37.5)	0.71
Neurologic disease *n* (%)	2 (15.4)	0 (0)	0.23
Cause for LTE *n* (%)			0.86
* Anoxic encephalopathy	6 (38.5)	3 (50)	
* Hemorrhagic CVA	5 (46.2)	4 (37.5)	
* Other causes	2 (15.4)	1 (12.5)	
Length of stay in ICU	10 (4.3–20)	7 (3–16)	0.09
Length of stay in Hospital	12 (4.3–20)	7 (3–17)	0.06
f-WIT (minutes)	19 (10.8–21.5)	14 (13–17)	0.09
TIT (minutes)	24 (17.5–27.5)	26 (19–33)	1
NRP (minutes)	132 (99.8–150.5)	103 (90–125)	0.15

* Clinical variables of patients enrolled in the clinical trial of donors in controlled donation after circulatory death (DCD). Qualitative variables were expressed as frequencies (%) and quantitative variables as median and interquartile range (25–75). IR, Interquartile range; CVA, Cerebrovascular Accident; HT, Hypertension; DM, Diabetes Mellitus; LTE, Limitation of Therapeutic Effort; F-WIT, Functional Warm Ischemia Time; TIT, Total Warm Ischemia Time; NRP, normothermic regional perfusion. *n* (%).

**Table 3 healthcare-10-00762-t003:** Quantification of malondialdehyde (MDA), at 60’ and 90’ after melatonin/placebo administration in DBD donors. IR, Interquartile Range.

Malondialdehyde (MDA), Median (IR)	Melatonin Group	Placebo Group	*p*
Difference at 0–60’	0.05 (−1.2; −0.6)	0.06 (−1.6; −0.9)	0.78
Difference at 0–90’	−0.17 (−1.9; −0.6)	−0.0083 (−1.9; −1.6)	0.55

**Table 4 healthcare-10-00762-t004:** Quantification of MDA from baseline level, at 60’ and 90’ after melatonin/placebo administration, and differences in concentration between 0–60’ and 0–90’ in cDCD (controlled Donation After Circulatory Death). IR, Interquartile range.

Malondialdehyde, Median (IR)	Melatonin Group	Placebo Group	*p*
Difference at 0–60’	−3.1 (−6.1; −0.8)	−1.2 (−3.1; −0.5)	0.21
Difference at 0–90’	−4.3 (−6.7; −2.6)	−1.6 (−2.9; −0.75)	0.004

## Data Availability

Not applicable.

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
