# Peer review of "Immunomodulation of Oxidative Stress during Organ Donation Process: Preliminary Results"

_healthcare, 2022, doi:10.3390/healthcare10050762_

Round 1

Reviewer 1 Report

The manuscript titled “Melatonin Treatment for The Immunomodulation of Oxidative Stress in Transplant Organs during Controlled Donation after Circulatory Death. Preliminary Results from A Randomized Clinical Trial” is an interesting work. However, there are some significant drawbacks in the manuscript.

These are few comments that need to be addressed: 

  1. The title is long and confusing. Rewording will be needed
  2. Abstract, lines 18-20, the objective of the work is not clear. Paraphrasing is required.
  3. Different font types in the abstract text should be avoided.
  4. Line 37, the word “organic” is not relevant
  5. Again, the last sentence (lines 73-75) of the introduction is not clear.
  6. Recipients’ and donors’ biological characteristics were not considered in analyzing the results or explaining the results in the discussion
  7. Mechanistic explanation of melatonin and a summary figure will be required
  8. The conclusion is only applicable to DCD but not DBD. Then also DCD is significant only at 0-90 but not 0-60. The conclusions are overstretched. The discussion and conclusion sections need to be updated/rewritten based on the results.
  9. Table 4 is missing from the text (though it’s there in the attached supplementary).
  10. Conclusions and experimental designs are not consistent. Also, factors associated with the effects of melatonin are not considered and discussed in the manuscript.

Reviewer 2 Report

Thank you for permitting me to review this manuscript 

please  better explain the difference  between DBD  and DCD

Line  51-54 please enumerate more molecules free radicals present in this process 

Line 70 please provide a reference (PPR)

Is MDA measurement a significant part of  oxyydative stress ? please develop

 Line 87 Please define SNG  

Fig 2 and table 3 the difference is not significant ? 

since the difference is not significant , why the autors claim a reduction in oxydative stress ? 

The authors should state assumptions and  conclsions strictly based on their findings  putative statements aare not justified when they are not supported by the results 

Round 2

Reviewer 1 Report

Authors have addressed the comments to a reasonable extent.

Author Response

We want to thank this reviewer for his/her positive and constructive comments on our manuscript, we believe our paper has increased its quality following his/her suggestions. 

Reviewer 2 Report

The authors have  improved the manuscript 

Fig 4 is useless since table 4 provides the same result and can be deleted

the authors have provided only one result significantly different for MDA 

A more cautious conclusion is needed 
